# Developing a Multilevel Decision Support Tool for Urban Mobility

Josep Maria Salanova [1,*], Georgia Ayfantopoulou [1], Evripidis Magkos [1,*], Ioannis Mallidis [1], Zisis Maleas [1], Santhanakrishnan Narayanan [2], Constantinos Antoniou [2], Athina Tympakianaki [3], Ignacio Martin [4] and Jenny Fajardo-Calderin [5]

1   Hellenic Institute of Transport, Centre for Research and Technology Hellas, 6th km Charilaou-Thermi Rd., 57001 Thessaloniki, Greece; gea@certh.gr (G.A.); imallidis@certh.gr (I.M.); zisismaleas@certh.gr (Z.M.)
2   TUM School of Engineering and Design, Technical University of Munich, Arcisstraße 21, 80333 München, Germany; santhanakrishnan.narayanan@tum.de (S.N.); c.antoniou@tum.de (C.A.)
3   Aimsun SLU, Ronda Universitat, 22 B, 08007 Barcelona, Spain; athina.tympakianaki@aimsun.com
4   Nommon Solutions and Technologies S.L., Plaza de Carlos Trias Bertrán 4, 2nd Floor, 28020 Madrid, Spain; ignacio.martin@nommon.es
5   Deusto Institute of Technology (DeustoTech), Faculty of Engineering, University of Deusto, 48007 Bilbao, Spain; fajardo.jenny@deusto.es
*   Correspondence: jose@certh.gr (J.M.S.); emagkos@hotmail.gr (E.M.)

**Abstract:** Decisions on transport policy measures have long-term and important impacts on the economy, environment and society. Transport policy measures can lock up capital for decades and cause manifold external effects. In order to allow policymakers to evaluate transport policies, the developed decision support tool facilitates the evaluation of the multidimensional impacts of the implementation of transport policies. The objective of the decision support toolset presented in this paper is to support transportation planning and design practices based on an integrated transportation analysis of the area of examination to determine the most applicable combination of mobility services. This paper provides a comprehensive description of the interactive decision support tool implemented to help cities and decision makers design their strategies and shape the urban mobility of the future.

**Keywords:** emerging urban mobility solutions; decision support tool

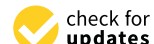



## 1. Introduction

Decisions on transport policy measures have long-term and important impacts on the economy, environment and society. In order to allow policymakers to evaluate transport policies, a decision support tool (DST) is required to evaluate the impacts of the implementation of transport policies. The described decision support tool (DST) in this paper is a scientific and technical procedure, aiming to identify the impact of potential urban mobility solutions, depending on the characteristics and mobility data of each city.

The DST presented in this paper was developed within the MOMENTUM project. The goal of the DST is to develop a set of new data analysis methods, transport models and planning support tools to capture the impact of these new transport options on the urban mobility ecosystem in order to support cities in the task of designing the right policy mix to exploit the full potential of these emerging mobility solutions. The developed DST is an online, user-friendly tool and consists of three levels. Based on the granularity of data, different scenarios for each level can be tested, as can be seen in Table 1. The cities participating are Thessaloniki, Madrid, Regensburg and Leuven. All cities presented a set of case studies with heterogeneous characteristics in terms of size, morphology, environmental, socioeconomic and cultural factors, mobility issues and policy goals.

**Table 1.** Input data and analysis for each level of the DST.

| | Input Data Requirements | Analysis Capabilities |
|---|---|---|
| **Level 1** | **Low:**<br>Demographics + socioeconomic data | **Analytical:**<br>Preliminary transportation design |
| **Level 2** | **Medium:**<br>Mobility data | **Extensive:**<br>Data-driven decision system |
| **Level 3** | **High:**<br>Full information using transport simulation tools | **Comprehensive:**<br>Transport planning |

Many European cities, especially small- and medium-sized cities, continue to use the traditional, strategic, four-step transport modeling approach for transport planning. This approach has been established and applied to various scenarios and, hence, has become the primary approach for travel demand modeling. The method assumes that the network input is dissected into traffic analysis zones, which contain relatively homogeneous socioeconomic factors and geographically close land-use objects (e.g., housing). There is an inertia to alter the traditional approach to other advanced methods due to insufficient data, a deficit of technical expertise and the convenience of simpler models. The advancement of the traditional four-step model, the agent-based approach, has been carried out at the level of individuals, as opposed to the level of the traffic analysis zones.

Decision Support Toolset

Figure 1 describes the flowchart of the implementation of the DST. Based on the available input data of each city described in Table 1, each city can achieve the most relevant level of detail. The first step is to discuss with the city and stakeholders the urban mobility services they need to investigate in order to develop the city's strategy. Afterwards, based on the available input transportation data and needs of each city, they can perform DST implementation. Level 1 provides a preliminary analysis of the examined area, indicating the most applicable modes for the city and the range of the efficient number of key factors of the services (e.g., stations, bicycles, scooters, buses). In Level 2, the DST provides a detailed investigation of the number of units and the locations of the stops. Level 3 includes a comprehensive transportation analysis of the area examined. For this level, current demand and supply are fed to the transportation model, investigating the impact of new emerging mobility solutions in the urban environment of the city.

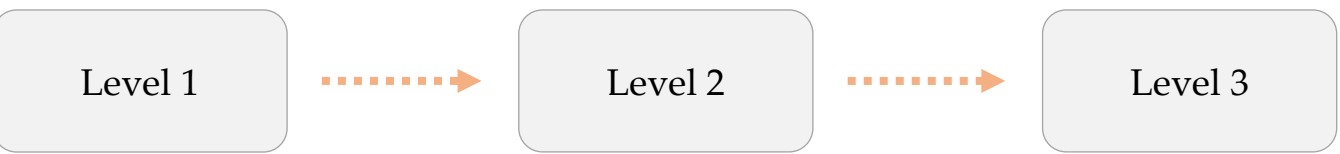

**Figure 1.** Flowchart of the DST.

Based on the implementation of the DST in the real cases where it was examined, we will present below three scenarios of the application of the DST in a city. In this set of scenarios, different approaches will be presented, based on the input data each city can provide.

Scenario 1

For this case study, we will assume that a city does not have mobility data or a transportation model. In this example, cities can only test Level 1. In this level, users need to add average values or highly aggregated socioeconomic, operational and functional variables. Based on the KPIs available for Level 1, users can identify the layout of their proposed solution, through produced dashboards, charts and values of the parameters (such as number of stations, docks, number of bicycles and scooters)

Scenario 2

For this case study, we will assume that a city has mobility data but not a transportation model. In this case, users can test until Level 2 of the DST. It goes without saying that users can test Level 1 but not Level 3, as a transport model is needed. In Level 2, users can use the algorithms available in the DST in order to convert data into the applicable format. Mobility data to be used as input data can be floating car data or OD matrices. Data such as bike lane networks, public transport lines and road networks are used as constraints in order to provide more accurate results. Based on the KPIs available for Level 2, users can identify more precise decisions compared to Level 1. Users can have access to a set of results such as the actual location of stops of the service, the fleet size needed and the capacity of stops and units (vehicles, bicycles and scooters). These values are critical parameters for the overall performance of the system under various scenarios.

Scenario 3

For this case study, we will assume that the city has a transportation model. In this case, a more analytical procedure needs to be followed, including a modal split of the available means of transport and optionally a synthetic population investigation. Due to the amount of calculation needed, these actions need to take place offline and then, using the results produced, need to be imported into the online version of the tool. Based on the KPIs available for Level 2, users can identify more precise decisions compared to previous levels. The mobility service simulator provides KPIs related to the users of each service. These include waiting times for each user to be served, travel times to complete their trip and number of served and unserved requests. Moreover, the model produced provides indicators regarding traffic emissions, car ownership and induced demand due to the introduction of new shared mobility services.

The added value of the DST is the unique option cities have to test and examine different scenarios, despite the amount of input transportation data they have. All cities can be used as test beds and perform a combination of levels provided by the DST. If applicable, cities are advised to test all levels in order to minimize the level of uncertainty for the produced results

Within the scope of this paper, we present a summary of the methodologies developed along with the connection to an online DST and the results from the investigation of a bike-sharing system in the city of Thessaloniki. In Section 2, the materials and methods investigated are described, while in Section 3, we present the results produced from the application of the DST in the city of Thessaloniki. In Section 4, the discussion and the added value of the produced results are described.

## 2. Materials and Methods

### 2.1. Scope of a Decision Support Toolset

The scope of the decision support systems has been changing through the years. Today, with the rampant advancements in information technologies, DSSs are used in a variety of applications across many domains. The ultimate goal of state-of-the-art decision support systems is to utilize the available data and implement the necessary models to help users in their decision making, both at the strategic and operational levels.

In general, a decision support tool or system consists of the following main components, described in Figure 2:

- A database management system (DBMS): This component holds the available data the DSS acts upon. Currently, the large amount of data collected and processed allows us to talk about Big Data.
- Models: These include the techniques, algorithms and processes, as well as the type of support provided and area of application. The current trend includes techniques derived from the popular artificial intelligence techniques and algorithms.
- User interface: This guides and helps the users through the decision-making process by providing a friendly, flexible, simple and interactive interface.

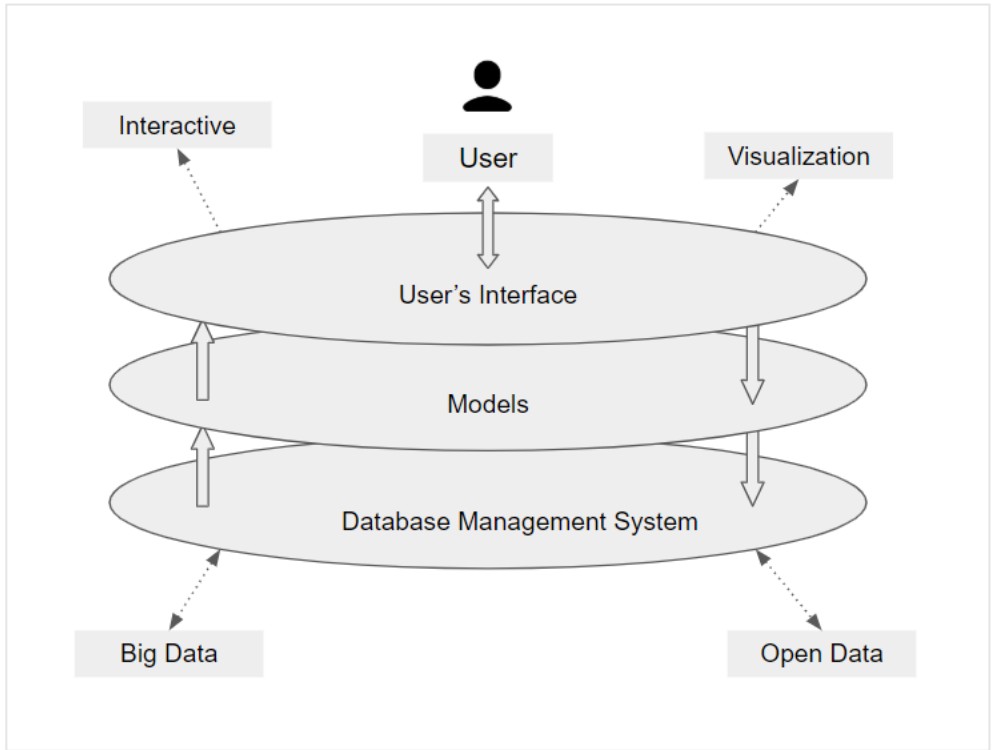

**Figure 2.** Components of a decision support tool.

Based on the main components described above and the descriptions of the previous sections, a high-level architecture diagram can be constructed.

As can be seen in Figure 3, the components of the decision support tool include the following:

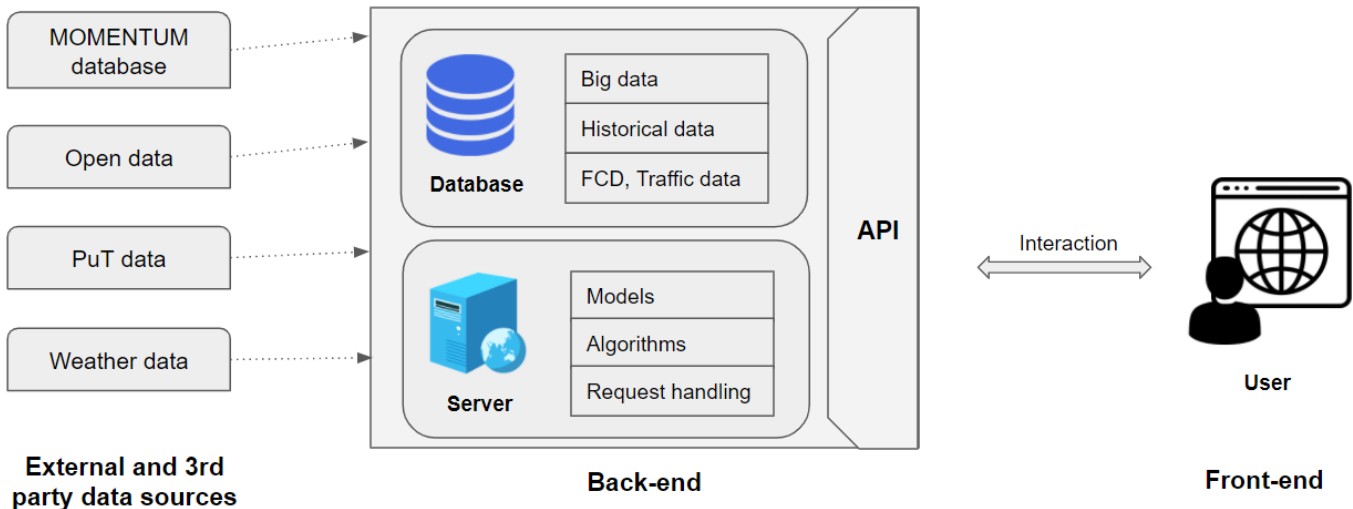

**Figure 3.** Architectural diagram of a DST.

Front-end: This component refers to the user's side of the system and includes the web platform, the interface the user interacts with and the minimal computations on the data retrieved from the server.

Back-end: This component provides the processing power of the decision support tool and includes the API used for the data exchange, the server used for the algorithms' execution and the database used for the storage of the data.

External data sources: This component feeds the back-end with the necessary data for the execution of the algorithms and is provided from external sources such as weather data.

### 2.2. Categories of Decision Support Toolsets

There are five different types of decision support systems described in the literature (Power, 2002 & 2004). The deviation among them is based on the way data have been received. The description of the different DST types can be found below:

Communication-driven decision support systems: The target group of these decision support systems is the internal teams, which include partners of an organization willing to establish an efficient collaboration, e.g., a successful meeting. A web or client server is the most common technology used to deploy these decision support systems.

Data-driven decision support systems: These systems are useful for querying a database or data warehouse to seek specific answers for specific purposes. They can be deployed using a mainframe system, client/server link or via the web.

Document-driven decision support systems: These systems are used to search web pages and find documents on a specific set of keywords or search terms. They can be implemented via the web or a client/server system.

Knowledge-driven decision support systems: These cover a broad range of paradigms in artificial intelligence to assist decision makers from different domains. Various data mining techniques are included, such as neural networks, fuzzy logic, evolutionary algorithm or case-based reasoning. Such techniques can be utilized for developing these systems to provide specialized expertise and information for specific decision-making problems. They can be deployed using client/server systems, the web or software running on standalone computers.

Model-driven decision support systems: These are complex systems developed based on some model (e.g., mathematical and analytical models) to help analyze decisions or choose between different alternatives. They can be deployed via software/hardware in standalone computers, client/server systems or the web.

Secondary components include the users themselves and visualization techniques and tools (e.g., geographic information systems (GISs)). The user's component includes the individuals or group of people that will use the DST, such as stakeholders, service providers, etc. The visualization component is very important for the user's experience while using the DST. Displaying the information in a compact and interesting way (e.g., through a map) can be beneficial for the overall success of the DST.

### 2.3. Developed DST and Produced KPIs

Within this section, we present the structure of the developed decision support toolset, the methodology followed for implanting the tool and the key performance indicators deriving from the tool.

Level 1—Preliminary transportation analysis

In the first level of the decision tool, an initial investigation of the urban mobility landscape of the city is examined. The preliminary analysis of this step requires a small amount of data such as geospatial socioeconomic data about the population of the studied area and the available operating fleets of mobility services. Due to the low granularity of input data, assumptions of demand, will be made so that decision makers can receive an initial estimation of potential urban mobility plans at a very low cost in terms of data.

Validation of the methodological procedure followed in Level 1 of the DST, was based on values cities provided. The input data received in order to validate Level 1 was twofold: on the one hand were associated with the values used for testing the DST while on the other hand, the expected or existing values of services (number of stations, docks per station, number of bicycles), currently operating the areas tested. Based on the validation of the values given and feedback received, the calibration of Level 1 was implemented.

However, based on the results expected by the cities, various mechanisms to "calibrate" the problem optimization were included in order to allow the cities to ask for more "social"

solutions and less economic. The features added to the tool are targeted at the expansion of possibilities to the city partners to assess the impact from the user's perspective or from the operational side. Furthermore, the addition of the sensitivity analysis of demand option in Level 1 aimed to give the user of the DST the ability to receive a range of solutions, not only the optimal values of the tool, in order to define the most applicable set of interventions to a specific city/area.

Level 2—Data-driven decision system

The aim of this level is to develop an analysis of the planning and evaluation of emerging mobility systems using a data-driven input data approach.

The added value of the Level 2 is based on the level of analysis that can be achieved, based on the available input data. The granularity of input data leads to the development of an in-depth analysis of the proposed modes of transport. To estimate the existing demand distribution, different algorithms are implemented in order to conclude efficient mobility service options. Information from various sources will be used in this procedure in order to provide the most accurate calculations. Information on the bicycle network, public transport stops and the road network will produce optimal solutions for the tested scenarios.

Level 2 provides data-driven information for the services tested, providing comprehensive operational and planning results. More information about the methodological approach followed in the Level 2 of the DST was based on the values cities provided. The input data were received in order to validate values used for testing the DST with expected or existing values of services currently operating in the areas tested. Based on the validation and the feedback received, features and methodologies for visualizing and presenting results were added to the tool. Finally, using available transportation information the city partners provided from data-driven information to OD matrices, scripts were produced and provided by the Centre for Research and Technology Hellas (CERTH) in order to produce the appropriate format of input data for Level 2 for all study cases.

Level 3—Comprehensive Transport planning

The last step of the multilevel decision support tool will involve a comprehensive analysis of the examined district by modeling the transport scheme of the selected city. Modeling for transport planning can be a powerful tool in understanding the potential traffic impacts of the proposed solutions if used in an appropriate way. It can also enable strategies to be developed, aiming to mitigate environmental impacts.

The interactive decision support dashboard, the policies under consideration and the KPIs are tailored to the specific requirements of each case study. More information about the KPIs to be produced can be found in D5.2 "Interaction Support Toolset". The implementation in the city of Thessaloniki worked as a testbed for the implementation in the rest of the cities. The new toolset is implemented on top of the existing mobility ecosystem, which holds a variety of data analysis and simulation tools, allowing the cooperation of public and private institutions. It has become a real-life conditions testbed for innovative mobility solutions.

The nature of the methodological stream of Level 3 is based on the investigation of the transport model of the city examined. In this level, comprehensive transportation information is used in order to examine the potential impact of emerging mobility services. The intermediate modeling approach [1] is used as the basis for Level 3. The structure of the modeling approach is briefly described in Appendix B.

On the online version of the DST, in Level 3, the user needs to proceed with the implementation offline and then use the visualization and analytical tools of the online version to assess the produced results

### 2.4. Methodological Approach

While the majority of cities continue to use the four-step modeling approach, modeling of shared mobility services calls for the advanced approach (i.e., agent-based approach), as modeling at the level of individual vehicles is a necessity for emerging mobility solutions.

Considering the aforementioned facts, there is a need for a pragmatic modeling approach, which adopts the disaggregate modeling principle from agent- and activity-based methods and integrates the same into the traditional strategic modeling approach. In this context, the project has developed an intermediate modeling approach and the implementation of a user-friendly DST in order to help users easily benefit from the knowledge gained. This method is located in between the traditional strategic and agent activity-based approaches, allowing to adequately model and evaluate the fleet planning and operations of shared mobility services.

Evidence-based policies with an impact on urban transport systems require reliable information about mobility patterns and travel demand. The traditional approach for the collection of travel demand information is based on surveys. These include household travel surveys and vehicle intercept surveys. Surveys provide rich information on mobility patterns and drivers' behavior. Nevertheless, they have limitations in terms of collecting and interpreting received data. Answers to the traditional surveys might be incorrect and imprecise, and they are expensive and time-consuming, which limits the size of the sample and the frequency of updates. This leads to many urban mobility plans being developed on the basis of information that does not precisely reflect the existing or future transportation environment. The potential of new, opportunistically collected data sources will help transport planners to overcome some of these limitations. These include sensor vehicular data, floating vehicular data, smart transport card data sensor personal data, floating personal data, social media data, mobile phone data, service operation data, etc. These alternative data sources can be fused with other contextual data sources (census information, land use, etc.) to complement and enrich the information obtained from traditional sources.

Data fusion and data analysis of conventional and opportunistically collected data sources can shed light on these issues. Within the implementation of the DST, we first performed a data collection and harmonization process. Then, using the harmonized data, we developed and validated different data fusion and artificial intelligence techniques to analyze mobility patterns and travel behavior. The adoption and usage patterns of some of the shared mobility modes available in the cities participating in the project were also studied.

For the operational part of the bike-sharing, micromobility and demand-responsive transport (DRT) service problems, most of the studies focus on the use of mixed-integer models along with dynamic programming and time-series algorithms. There are studies that develop models for the redistribution of bikes across stations with the use of combinatorial optimization and corresponding heuristics [2,3]. Related to this problem, inventory control theory is another approach used in the literature to solve related problems [4]. Additionally, demand prediction is also considered an important task for managing micromobility operations [5,6]. These initiatives are used also in the developed DST, while one of the objectives is to establish a system that allows resilient and sustainable operations management of such services. In the same way, the operational aspect of DRT services is well solved and studied, in comparison to the planning. The dial-a-ride problem formulation is the most common approach to optimize the routes and accurately serve the demand that services achieve to arrange the routes and accurately serve the demand [7].

Strategic planning and facility location are also core problems of the services related to the DST. Most of the studies that handle facility location and demand coverage planning problems use as input some possible candidate stops and try via linear-integer programs to define the best subset given some service-level constraints [8,9]. Similarly, mixed-integer algorithms and two-stage formulation have been examined in [10] to design a DRT service network. On the other hand, some studies make use of analytical statistical models that estimate the socioeconomic optimum of the fleet size required to serve the demand of an area [11]. In our solution, the socioeconomic approach is used to obtain an estimation of the optimal values, while the facility location, fleet size and system capacity are designed based on real distributions of the spatial demand. Moreover, as the dynamics of the system

evolve over time, it is important to model this process in our computations according to the system dynamics paradigm [12]. The facility location is determined for all the services under a general-purpose *p*-median model [13], based on candidates produced by a variant of the k-means algorithm [14,15]. Lastly, the fleet size, the capacity of vehicles and the number of docks are calculated via guided optimization models [16,17] and discrete event-based evaluation of each scenario. Searching over the literature, there is a vast palette of techniques for optimizing the different steps of a complete study, including data collection, data processing/cleaning, pattern recognition, decision analytics and optimization and robust optimization with simulation. The novelty of this study lies in the presentation of a unified process that streams this pipeline of operations, starting from data collection until the final decision about the optimal system parameters, packed in a user-friendly tool that improves the intuition and comprehension of public authorities such as city councils and transport consultants.

### 2.5. Key Performance Indicators (KPIs)

Performance measurement is a useful tool that can help decision makers and authorities to assess the importance of transportation and appropriate investment in transportation investments. The ultimate purpose of measuring performance is to improve transportation services for users. Moreover, performance measurement provides both important inputs for setting priorities and critical information that helps decision makers detect potential problems and make corrections in order to meet the goals and objectives of future mobility.

In the development of the proposed DST, different scenarios and transport modes are tested in order to specify the transport system dynamics. Network performance indicators are essential parameters in transport modeling evaluation models. The development of the DST and the research activities encompass the development of assessing the impact of new mobility services in the test case cities of Thessaloniki, Leuven, Madrid and Regensburg. Questions and needs related to urban mobility services in the cities examined were used in order to form the produced KPIs. The test cases of the four cities are:

- Scenarios and policies for each city;
- Datasets available in each city;
- Novelties and evolutions;
- Mobility policy priorities;
- Suggested policies to be tested;
- Questions to be addressed.

Considering the mobility policies and questions to be addressed for each case study, a list of KPIs was compiled. It is important to mention that, due to the different levels of granularity of input data for each level of the DST, the type of KPIs that can be derived varies depending on the data availability. Table 2 below shows the KPIs available in the DST for each examined Level of the tool.

**Table 2.** List with available KPIs for every level.

| KPIs | Level 1 | Level 2 | Level 3 |
|---|---|---|---|
| System's cost for each scenario | Provided | Provided | Provided |
| Travel time | Provided | Provided | Provided |
| Kilometers traveled | Provided | Provided | Provided |
| Number of units needed (vehicles, bicycles, etc.) | Provided | Provided | Provided |
| Passengers' waiting time | Provided | Provided | Provided |
| Demand coverage | | Provided | Provided |
| Accessibility | | Provided | Provided |
| Network coverage | | Provided | Provided |

**Table 2.** *Cont.*

| KPIs | Level 1 | Level 2 | Level 3 |
|---|---|---|---|
| Fleet's management operation pick-up points for DRT, station's locations from BS and micromobilitiy | | Provided | Provided |
| Modal split (BS, CS, RS, conventional systems) | | | Provided |
| Kilometers traveled per mode | | | Provided |
| Network's performance indicator (congestion, traffic flow, delays, travel times, queue lengths) | | | Provided |
| Use of active mobility means | | | Provided |
| Usage rate for each rate (number of trips, percentage of time used) | | | Provided |
| Car ownership (number of people per 1000 citizens) | | | Provided |

## 3. Results

In this section, the results of the implementation of the decision support toolset are presented. The city of Thessaloniki was used as a test bed. All three levels of the DST are described below, along with the input and output data of the tested scenarios.

Level 1—Preliminary transportation analysis

Thessaloniki's case study focused on how a DRT service should be implemented to contribute to sustainable mobility in the city, the role of ridesharing in the transport system of the city and the impacts of bike sharing and micromobility in transport planning. Socioeconomic and functional variables' characteristics were used as input data for Level 1 implementation, calculated as part of the Sustainable Urban Mobility Planning (SUMP) process for the city of Thessaloniki. The implementation of Level 1 was performed based on the recently updated and calculated values for the city of Thessaloniki. Socioeconomic and functional variables' data, such as the demand for services and value of time, were calculated using a stated preference study in the city of Thessaloniki for the purpose of preparing the new SUMP for the city. Values such as the cost of operation were calculated based on the operational costs of existing systems in the city.

For the bike-sharing service in Thessaloniki, different scenarios were tested. The different scenarios included changes in the operational, socioeconomic variables and estimated demand for bicycle trips in the examined area. The weight assigned for the cost of the user (weight refers to a value selected by the user of the DST in order to define the level of influence of the cost) was tested in the values between 0.1 and 1 in order to investigate the impact of the cost of using the provided service to the users. For the tests included in the Annex, tests with a weight of 0.5 are included, as these values were calculated to be more associated with the case study of Thessaloniki. Furthermore, the maximum waiting time was tested for values ranging from 1 to 6 min, while the maximum walking time the users need to walk to reach a station fluctuated from 5 to 10 min. Decision variables, including minimum and maximum values for the number of stations and docks, were selected based on the current number of stations and units operating in the city. In Thessaloniki, the available bike-sharing system is not sufficient to cover the demand of users. Hence, different scenarios were tested in order to identify the best solution. Finally, the demand range that was tested varied between 80 and 120 percent of the demand.

In this example, one scenario for the bike-sharing system is presented. The values used for the scenario of the bike-sharing system are described in Table 3. In the images below, the produced results are presented (Figure 4). The values used for the various tests examined for Level 1 for all available services are included in the Appendix A.

**Table 3.** Input values used for Level 1.

| Service | Bike Sharing |
|---|---|
| Country | Greece |
| City | Thessaloniki |
| Population | 1,012,297 |
| Square meters of area of interest | 19.3 km |
| **Cost of operation** | |
| Bicycle operating cost per km | 0.1 EUR/km |
| Bicycle depreciation cost per hour | 0.1 EUR/km |
| Operator cost per hour (per station) | 0.2 EUR/km |
| Dock depreciation cost per hour | 0.05 EUR/km |
| Weight assigned to the user (range from 0 to 1) | 0.5 |
| **Socioeconomic and functional variables** | |
| Value of time of users | 15 EUR/km |
| User walking speed | 5 km/h |
| Bicycles travel speed | 12 km/h |
| Mean demand of the area | 50 trips/h |
| Standard deviation of demand of the area | 8 trips/h |
| Travel time activation | Yes |
| **Constrains** | |
| Maximum waiting time | 3 min |
| Maximum walking time | 8 min |
| Decision Variables | |
| **Number of stations** | |
| Min | 1 |
| Max | 100 |
| **Number of Docks** | |
| Min | 1 |
| Max | 50 |
| Run sensitivity module for demand | Yes |
| **Demand range around the one declared above (%)** | |
| Min | 80 |
| Max | 120 |

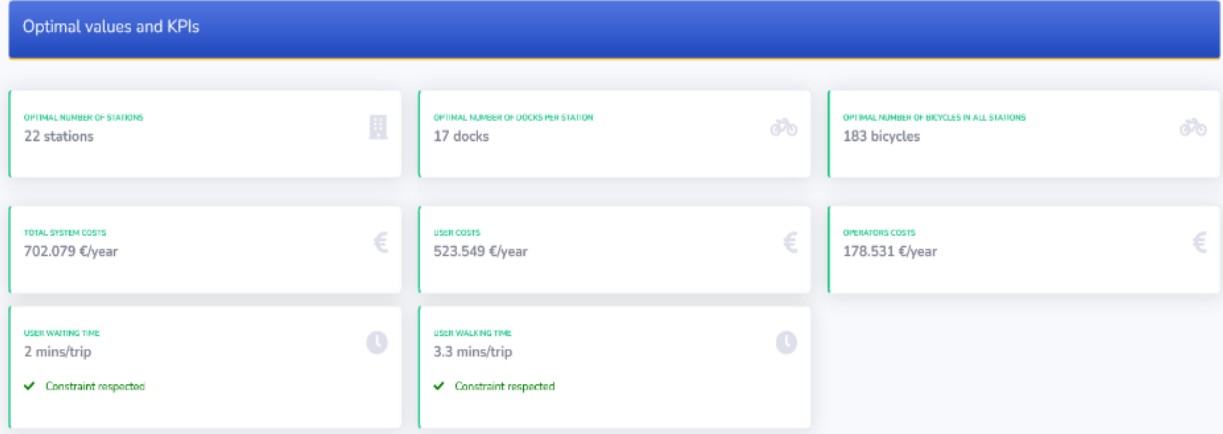

**Figure 4.** Produced KPIs.

The first set of results produced by Level 1 of the DST includes optimal values and KPIs based on the values used as input data. The number of bicycles and docks is presented along with user and operator costs in Figure 5.

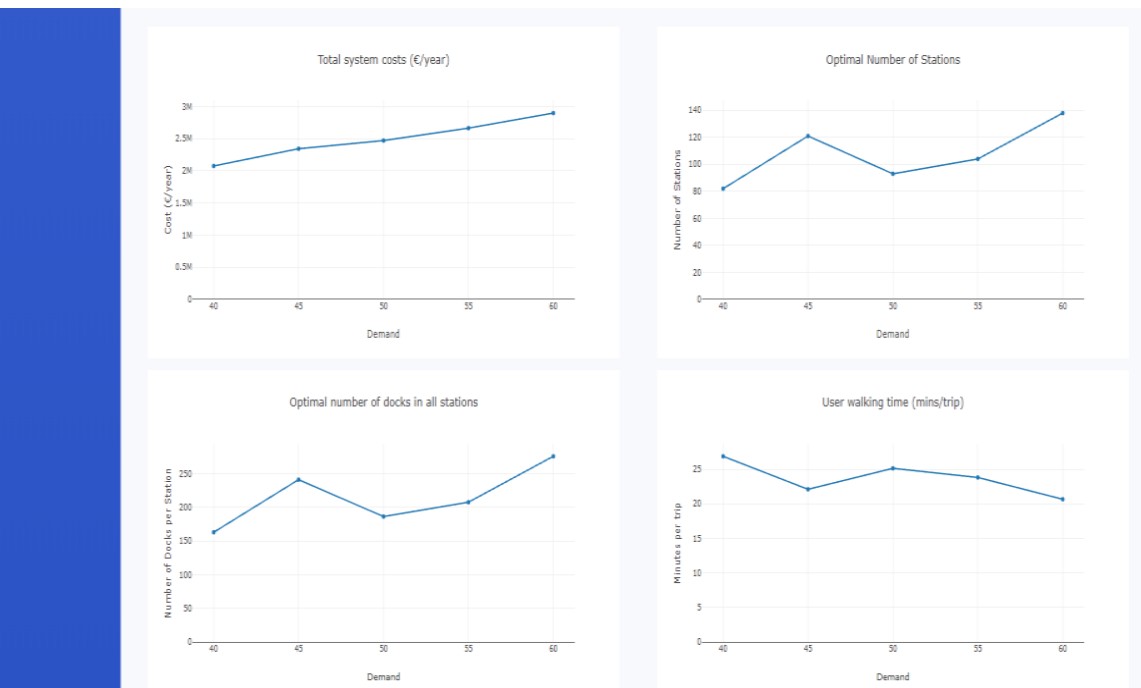

**Figure 5.** Range of optimal values.

The following charts present the calculated range of optimal solutions for the different indicators in the bike-sharing system in Thessaloniki. Based on Figure 6 the optimal number of stations can range from 9 to 60 (and maybe more), while the number of docks in each station depends on this value. The solution should be in the light-blue area. This is a useful tool in order to restrict the search range in Level 2 rather than searching in large state spaces, which can lead to high computational effort.

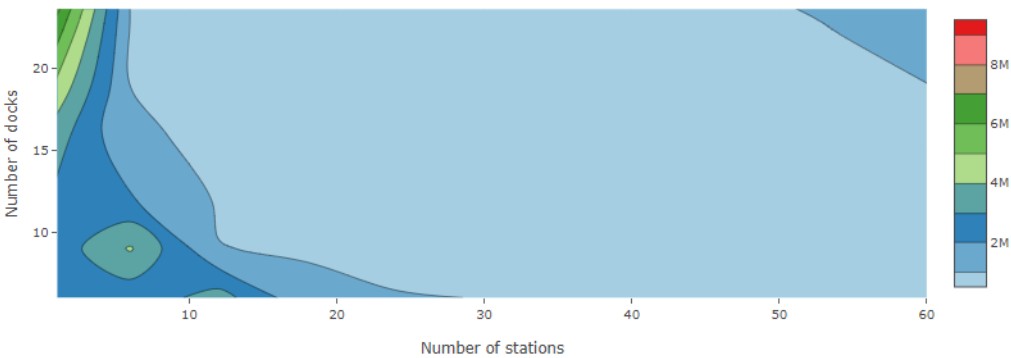

**Figure 6.** Impact of number of stations on the potential number of docks.

Level 2—Data-driven decision system

Level 2 extends the uniform distribution assumption and considers the data of spatially distributed trips. The analysis uses a dataset of real trips produced by operations of a scooter-sharing service in the area of Thessaloniki. An example of the distribution of scooter trips is depicted in Figure 7. The tool illustrates the distribution for each time frame available in the dataset of demand. The geographical representation of the demand helps the user to understand the demand patterns and better interpret the whole process including the results.

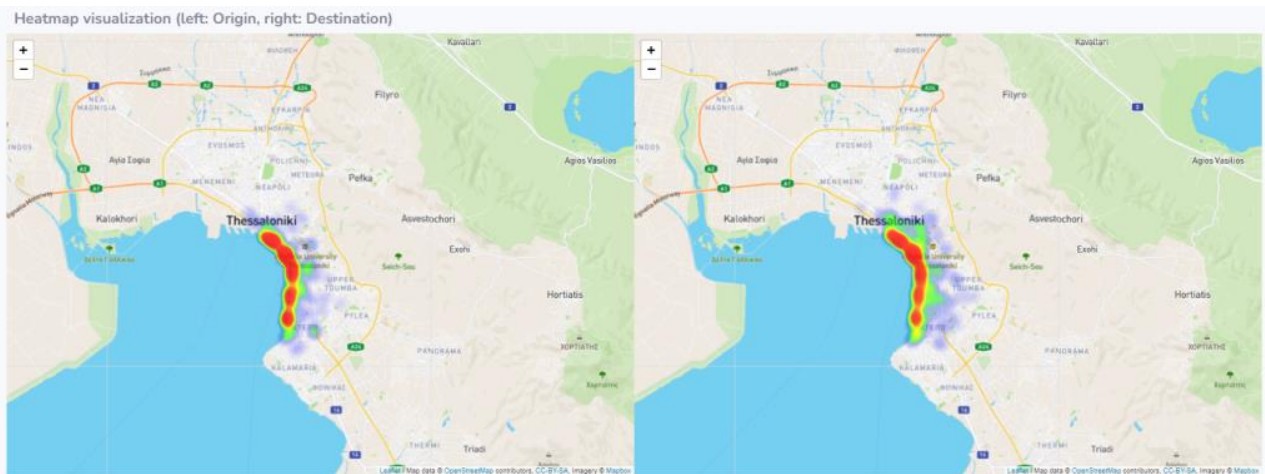

**Figure 7.** HeatMap visualization of the demand.

The analysis also takes into account the bike lanes network of Thessaloniki as the tool offers that option. According to this distribution and the desired goal of a 300 m (on average) walking distance per passenger, a set of station 130 candidates is generated (Figure 8). The distribution of the candidate stations in the area of study is proportional to the spatial distribution of the demand. The aim is to adopt the demand patterns in this set to better fit the demand. Intuitively, the areas with higher demand may also require a higher density of stations that ensure a reliable service level, along with higher proximity rates.

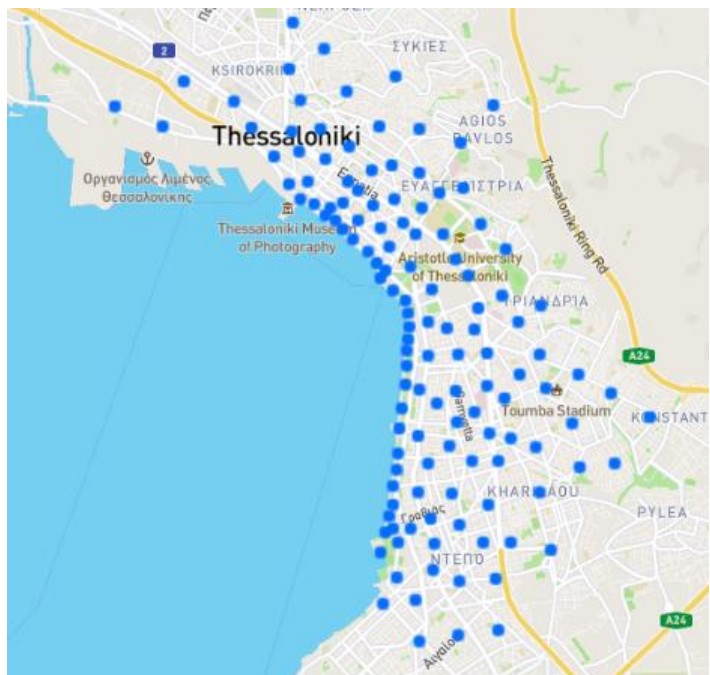

**Figure 8.** Station candidates according to DST.

Then, the facility location module carries the computation to define the final set of stations and the related number of docks each one should have. The final set of stations minimizes the walking distance of the origin and the destination of each trip. Additionally, the subset of stations that were finally selected is closer to the bike lane network. Small adjustments to the final setup are established through manual allocation to find appropriate urban spaces that do not disturb the urban infrastructures. The optimal bike-sharing

network is presented in Figure 9. The capacity and the inventory of each station were also calculated via heuristics, which were developed especially for the purposes of Level 2.

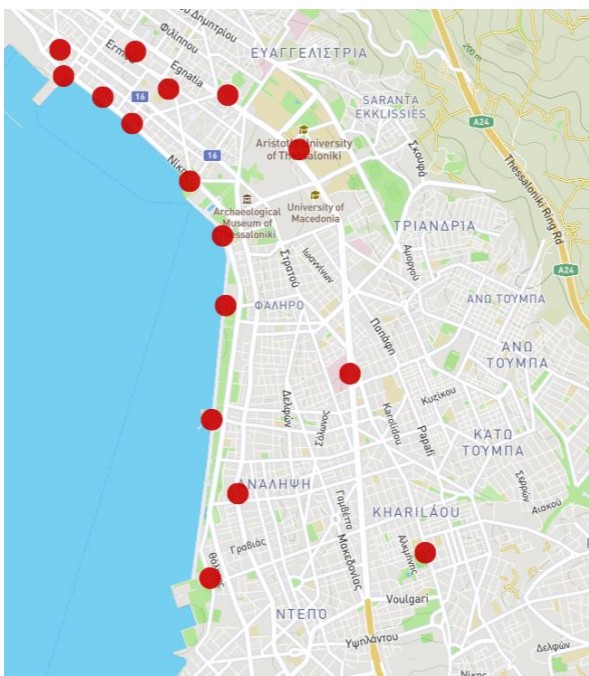

**Figure 9.** Bike-sharing network.

The calculation of the size of each station is characterized by arrival/departure rates that affect the queuing properties [18] of each one. Based on these rates, a Monte Carlo simulation was performed to fine-tune the capacity and the inventory in each hub. To examine the optimal strategy of setting the size, a series of experiments were conducted. Experiments S1 and S2 consider the uniform inventory and available parking space in each station equal to 10 bike/20 docks and 15 bike/30 docks, respectively. In the S3 series, the size and inventory should be proportional to the difference between the arrival and departure rates of each station. Therefore, stations having a higher arrival rate than the departure offer more available space (capacity inventory), while the opposite requires higher inventories and limited available capacity.

Strategies 3 (S3) and 2 (S2) do not present any statistically important differences. However, S2 requires fewer investments in bikes and docks with nearly the same performance. Based on the results illustrated in Figures 10 and 11, the rebalancing operation should take place every 12 h (720 min) at most. To achieve this, only one vehicle is needed according to the last step of Level 2 of the DST. The vehicle should group the stations into two partitions and serve them separately to get this operation more efficiently. This operation was simulated with the use of the Aimsun Ride toolset.

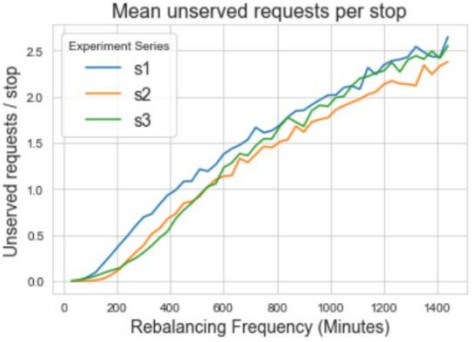

**Figure 10.** Mean of unserved requests per stop.

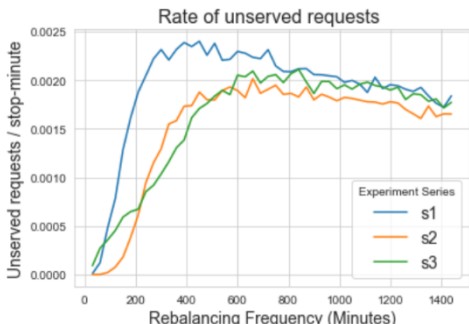

**Figure 11.** Rate of unserved requests.

In Figure 12 below, an example is illustrated where the user can see the visualized routes of the requests served. The green figures depict the origin of the request, while the red figures depict the destination points.

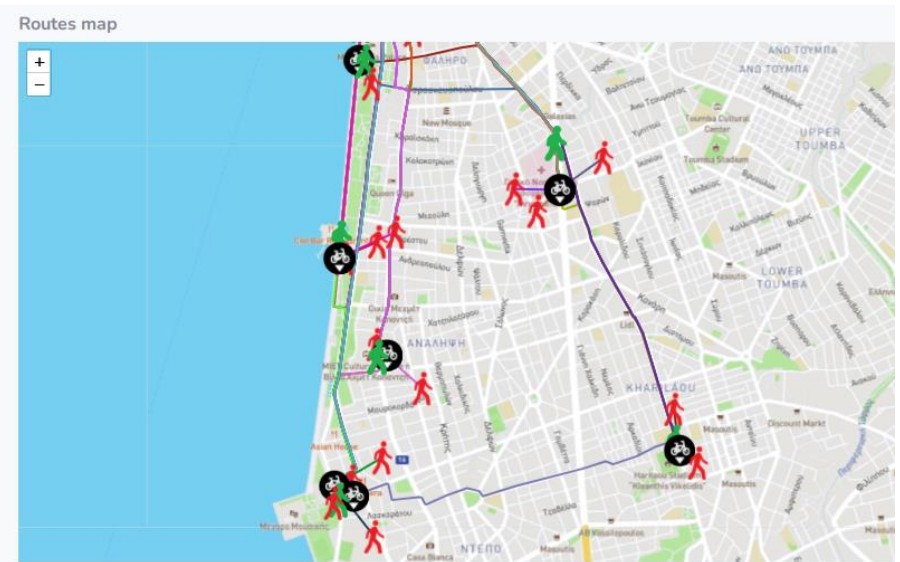

**Figure 12.** Locations of bike stations. Origins and destinations of the requests.

Level 3—Comprehensive Transport Planning

One of the scenarios tested for the case of Thessaloniki was to evaluate the introduction of the extensive bike-sharing system in the city. Currently, a station-based bike-sharing system and a floating bike-sharing system are operating in the city. The aim of this scenario is to enrich the existing service and produce insights into expanding the use of the service.

The existing VISUM transport model in the city is used as the basis for this case study. The existing model was originally developed in VISUM 15.0, and the most recent update of this model took place as part of the Sustainable Urban Mobility Plan of the Municipality of Thessaloniki (2017–2019).

A base synthetic population was generated using the open-source software PoPGen. Synthetic individuals were generated based on data from a stated preference (SP) survey (collected in the year 2017, with a sample of around 11,000 participants), along with aggregate demographic statistics obtained from the Hellenic Statistical Authority for the extended region of Thessaloniki. Socioeconomic data used include household size, car ownership level and income at the household level, along with the location zone of the household. At the level of individuals, utilized variables include age, gender, education and employment status. The levels for categorical variables are selected based on the specifications of the disaggregate mode choice model, which is applied in the subsequent step of the modeling framework.

To estimate the modal split for the bike-sharing system, a disaggregate mode choice model developed within the project [19] is utilized by adopting the utility specification corresponding to this service. The variables used in the mode choice model, along with their coefficient values and description, are presented in Table 4.

**Table 4.** Variables used in the mode choice model.

| Name | Coefficient | Description |
|---|---|---|
| Age_20_To_44 | 1.444 | 1 if the age of the individual is between 20 and 44, 0 else. |
| Male | 1.438 | 1 if the individual is male, 0 else. |
| University_Or_Vocational_Degree | 0.922 | 1 if the individual has university or vocational degree. |
| has_PTPass | 1.071 | 1 if the individual has public transport pass, 0 else. |
| Bikesharing_Supply | 1.337 | The inventory of available bikes in the nearest station (the average inventory used as an estimator of the related variable). |
| Trip_Dist_KM_2To5 | 2.369 | 1 if the individual's travel distance is between 2 to 5 km, 0 else. |
| Trip_Dist_KM2 | 1.569 | 1 if the individual's travel distance is less than 2 km, 0 else. |
| Travel_Time_Mins15_Bikesharing | 1.258 | 1 if the individual's travel duration is between 0 to 15 min, 0 else. |
| Travel_Time_Mins30_Bikesharing | 0.792 | 1 if the individual's travel duration is less than 30 min, 0 else. |
| Vehicle_Non_Availability_Bikesharing | −100 | 1 if there is no availability of the bike-sharing service for a particular trip, 0 else. |

Furthermore, the estimated disaggregate demand for the shared systems, along with data and information related to the request characteristics, service and network supply derived from Level 2, are fed into the fleet management model to optimize the trip plan solutions and simulate the operations to serve the demand for the shared services. The Aimsun Ride shared mobility service simulation platform, implemented within the Aimsun Next [20] transport modeling simulation software, is used to obtain precise and accurate information about transport network behavior, while fleet management algorithms are used to handle both planning and operational components in the simulation experiments. For the bike-sharing system in Thessaloniki, the number of requests for the service was 75 for the peak hour examined (08:00–09:00). In the image below, you can see the locations of the requests. Table 5 presents the KPIs from the implementation and simulation of the bike-sharing system in the area examined in Thessaloniki.

**Table 5.** Key performance indicators produced.

| KPIs | Values |
|---|---|
| Requests served | 100% |
| Number of requests completed | 75 |
| Number of bicycles used | 73 |
| Average total trip (min) | 16.7 |
| Max total travel trip (min) | 27.3 |
| Min total travel trip (min) | 8.1 |
| Average "bike" travel time (min) | 9.5 |
| Max "bike" travel time (min) | 20.5 |
| Min "bike" travel time (min) | 3.2 |
| Average walking time (min) | 7.2 |
| Max walking time (min) | 12.3 |
| Min walking time (min) | 2.2 |
| Average traveled distance (km) | 4.5 |
| Max traveled distance (km) | 8.5 |
| Min traveled distance (km) | 2.5 |

As can be seen, all requests assigned were served. The average total trip is 16.7 min. The term total trip refers to the sum of walking and "bike travel" time. This includes the time the user needs to walk from their destination to the bike station, ride their bicycle to

the final bicycle station and walk to their final destination. The average "bike travel" time is 9.5 min, while the average walking time is 7.2 min.

In the figure below (Figure 13), the user can have a more detailed analysis of the produced routes. For each route, various KPIs produced are described, such as the travel time for each segment and the walking time of the users of the service.

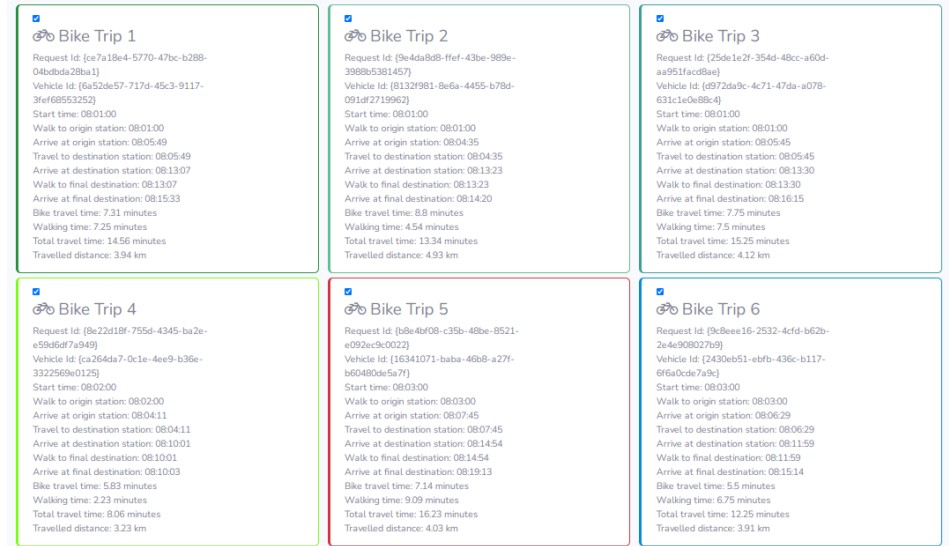

**Figure 13.** Information on every trip for the bike-sharing system.

## 4. Discussion

Decisions on transport policy measures have long-term impacts on society. Transport policy measures can lock up capital for decades and cause manifold external effects. Due to the growth in the urban population, there has been an increase in demand for mobility and, consequently, an increase in the number of vehicles on the roads. The increased levels of traffic congestion indicate a strong and imminent need for cities to foster sustainable and eco-friendly solutions for urban mobility. In order to allow European policymakers to evaluate transport policies, a decision support tool (DST) is required to evaluate the economic, environmental and social impacts of the implementation of transport policies.

In the DST proposed, each level entails a different degree of complexity, both in the input and in the output data. The proposed three-level DST is a scientific and technical procedure aiming to explore the available urban mobility solutions for each examined area, depending on the characteristics (socioeconomic, spatial, existing infrastructure, etc.) of each case study. In each stage of the decision support tool, a different level of detail is followed depending on the availability of the input data.

Level 1 is an optimization process, providing the optimum range of results (docks, number of units, etc.). However, based on the results expected by the cities, various mechanisms to "calibrate" the way the problem is optimized were included in order to allow the cities to ask for more "social" economic solutions. Features added to the tool targeted the expansion of possibilities to the city partners to optimize the supply layout and evaluate the operational system of the services. Furthermore, the addition of the sensitivity analysis of demand option in Level 1 aimed to give the user of the DST the ability to receive a range of solutions, not only the optimal values of the tool in order to define the most applicable set of interventions.

Level 2 aims to embody methods that utilize spatially distributed data from trips or OD matrices to perform strategic decisions for the service. The user of the tool should define the desired characteristics of the service to let the algorithm decide the resources needed to fulfill the requirements. In this step, the user should tune the values of demand in the examined area, road network and constraints such as bicycle network and public transport

stops. Based on these criteria, the planning module can return the optimal number and location of stops/docks, the capacity of vehicles or stops and similar system parameters. The operational module helps to evaluate each strategic setup based on performance metrics. These metrics are also used in the optimization of planning parameters, as they reveal possible surpluses or shortages of resources for the service.

Level 3 of the DST involves a comprehensive analysis of the examined district by modeling the transport scheme of the selected city. The fleet operational algorithms developed for each examined service, synthetic population integration, along with the mode choice model developed, interact with the shared mobility services simulation platform, Aimsun Ride. This integration allows one to execute the requests in a simulation environment according to the optimized trip plans. Various indicators with respect to both the users of the service, as well as the service performance, are obtained. The main advantage of the simulation is the capability to provide more accurate predictions and extended KPIs, which are not available in Level 1 and Level 2 of the decision support tool due to the nature and limitations of the methodologies that they utilize. The proposed modeling schema consists of various models and algorithms that, when integrated, can provide the necessary functionalities in order to perform more accurate and realistic strategic planning and evaluation of emerging shared mobility services.

New emerging mobility services have recently been introduced in the urban environment of many cities across the world. These new sustainable transportation solutions demand a different approach in terms of designing the urban environment of the cities. Agent-based models for emerging urban mobility services developed provide a new transport approach to the existing traditional methods, with more updated and compliant input data. The examined decision support toolset presented in this paper is a powerful tool for decision makers to coordinate their actions in adopting emerging mobility solutions by testing various scenarios for emerging urban mobility scenarios. The DST includes the investigation of new emerging mobility services in the existing transportation environment of a city, utilizing various sources of input data. Algorithms and methodologies developed in the DST interact with state-of-the-art transport software, providing comprehensive network performance indicators.

Based on the cutting-edge scientific and technological innovations developed in the presented decision support toolset, it can be used for future research to improve the capabilities of the tool by including more emerging mobility services, upgrading the state-of-the-art open-access platform aiming to transform the future of transport and mobility systems.

**Author Contributions:** G.A.: Data Curation, Project administration, resources, supervision, validation, J.M.S.: Data Curation, Project administration, supervision, validation, conceptualization, E.M.: formal analysis, investigation, methodology, visualization, writing—review& editing, Z.M.: formal analysis, investigation, visualization, writing—review& editing, writing-original draft, I.M. (Ioannis Malidis): formal analysis, investigation, visualization, writing—review& editing, writing-original draft, S.N.: formal analysis, investigation, visualization, writing—review& editing, writing-original draft, C.A.: Data Curation, Project administration, supervision, validation, conceptualization, A.T.: Data Curation, Project administration, supervision, validation, conceptualization, I.M. (Ignacio Martin): Data Curation, Project administration, supervision, validation, conceptualization, J.F.-C.: formal analysis, investigation, visualization, writing—review& editing, writing-original draft. All authors have read and agreed to the published version of the manuscript.

**Funding:** This project has received funding from the European Union's Horizon 2020 research and innovation program under Grant Agreement No. 815069 (Project MOMENTUM) and the European Union's Horizon 2020 research and innovation program under Grant Agreement No. 774199 (Project IRIS).

**Data Availability Statement:** All developed algorithms and datasets are incuded in MOMENTUM's repository. For confidentiality datasets are not available in external partners of the project. More details about the projects and the methods followed, we refer reader to deliverables of the project. (https://h2020-momentum.eu/, accessed on 29 March 2022).

**Conflicts of Interest:** The authors declare no conflict of interest.

## Appendix A

**Table A1.** Values used for testing the bike sharing system.

| | | Service: Bike-Sharing System | | | | | | | | | | | | | |
| Area of Interest | | Cost of Operation | | | | | Socioeconomic and Functional Variables | | | | | | Constraints | | Decision Variables | |
| City | Area | Bicycle Operating Cost per km | Bicycle Depreciation Cost per Hour | Operator Cost per Hour (per Station) | Dock Depreciation Cost per Hour | Weight Assigned to the User | Value of Time of Users | User Walking Speed | Bicycles Travel Speed | Mean Demand of the Area | Standard Deviation of Demand of the Area | Travel Time Activation | Maximum Waiting Time | Maximum Walking Time | Number of Stations | Number of Docks |
|---|---|---|---|---|---|---|---|---|---|---|---|---|---|---|---|---|
| **Thessaloniki** | | 0.1 | 0.1 | 1 | 1 | 0.5 | 6 | 5 | 12 | 50 | 8 | YES | 3 | 8 | 1 to 21 | 1 to 40 |
| **Thessaloniki** | | 0.3 | 0.1 | 1 | 1 | 0.5 | 6 | 5 | 12 | 50 | 8 | YES | 3 | 8 | 1 to 21 | 1 to 40 |
| **Thessaloniki** | | 0.1 | 0.1 | 1 | 1 | 0.5 | 6 | 5 | 12 | 50 | 8 | YES | 6 | 8 | 1 to 10 | 1 to 40 |
| **Thessaloniki** | | 0.1 | 0.1 | 1 | 1 | 0.5 | 6 | 5 | 12 | 50 | 8 | YES | 3 | 8 | 1 to 20 | 1 to 15 |

**Appendix B**

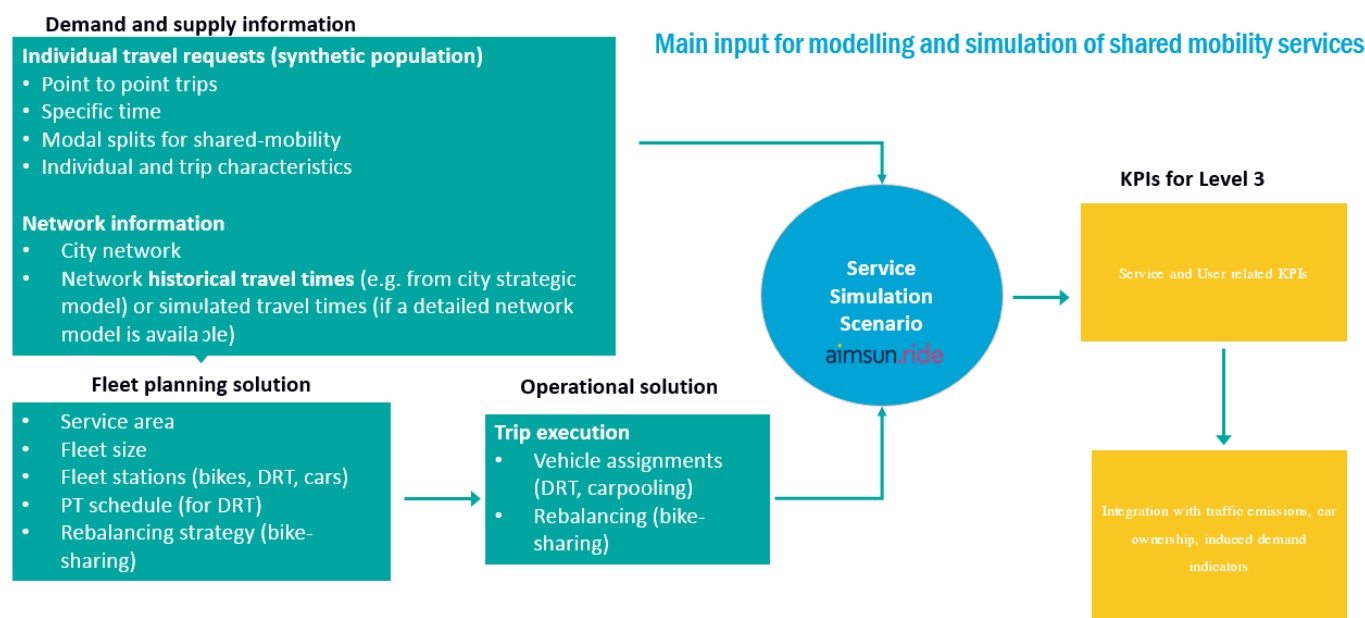

**Figure A1.** Explainatory digram of level 3 of the DST.

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
