# Peer review of "Developing a Multilevel Decision Support Tool for Urban Mobility"

_sustainability, doi:10.3390/su14137764_

Round 1
Reviewer 1 Report
The paper entitled Developing a multilevel Decision Support Tool for urban mobility proposed a comprehensive description of the interactive Decision Support Tool implemented to help cities and decision-makers to design their strategies and shape the urban mobility of the future. Although it is interesting, the authors failed to present the study.
- The introduction is not written well. The introduction should explain the background of the problem, pointing out the main goal of the paper and the main scientific contribution of the previous studies. The current form is lack that information.
- I advise the authors to provide a comprehensive literature review, to support section 2. The current literature used is not adequate and has many flaws. It does not provide insight into the research gaps the paper is trying to fill. It is not clear which are the paper's main contributions to the previously studied literature.
- The methodology section should be fully described. Provide more relationships and explanations about method innovation. The current form seems to jump from one section to another without a clear connection. Maybe providing a methodology flowchart would help the readers to understand the paper clearly.
- The explanation of each KPI level is unclear. Where did the authors come up with those KPIs? literature or expert judgment?
- KPI level 1 is stated to be a result of the optimization process? Why it is not clearly explained in the paper?
- Authors should show the decision tools' overall flow, probably by providing a flowchart? or the framework to show how the decision tools work.
- Name of the variable in table 2 are unclear, authors should explain each of the variables or use the actual variable as the name.
- There are many things that are not clearly explained in this paper. The result and discussion, also the conclusion are short and do not answer the research questions (which are also not clearly stated).
- The case study is too narrow and specific to be used for drawing some general conclusions. Future research directions are needed. The authors should have given at least 3-to 5 solid future research directions which would be interesting to the majority of the readership of the journal.
- the authors also failed to follow this journal's guidelines.
Reviewer 2 Report
The paper presents an interesting decision support tool that can be used to evaluate different transportation projects according to differentia criteria. The paper has some merit and provides a tool that could be useful to decision makers. My main comment is that I think the Introduction and Methods sections should be re-written to better explain the purpose of the project and the details of the tool. Parts of the Methods section are vague, confusing, or lacking in detail. Additional comments are below:
- Page 1, line 35: "on the….on the availability of input data." This seems like a mistake.
- The Introduction would be more effective if it described the impacts that the tool measures. The Introduction could also explain the contribution that this study makes to the literature.
- Line 52: Define DRT.
- Line 58: Unnecessary question mark.
- Page 2, Methodological approach: Before you start citing all these studies, you need to make it clear what exactly your tool does. What are the components of your tool, and how do these studies relate to what you are doing? Otherwise, I am confused as to why you are citing these studies and how they are related to your study. Start off with a broad description before getting into the details, and include sentences that tie everything together. This section feels very random and disorganized.
- Line 113-114: "Depending on the applicable urban mobility scheme, the appropriate analytical approach will be implemented." What does this mean? What are the urban mobility schemes? What are the analytical approaches? And how do you determine what is appropriate?
- Line 124-126: "It is important to mention that Level 1 is an optimization process, so it may not provide the real values, but the optimum based on the mathematical formulations and the hypothesis." I'm confused. What is the optimization model? What are the mathematical formulations or hypothesis?
- Line 140-141: "the estimation of the supply of the existing demand distribution" This does not make any sense.
- Line 154: Define CERTH
- The Methods section does not provide enough detail about the how the tool actually works.
- Line 189: Define SUMP.
- Line 198-200 "Weight assigned for the cost of the user was tested in the values between 0, 1 and 1, in order to investigate the impact of the cost of using the provided service to the users." What does this mean?
- How can decision makers actually use this tool? Can they access it and implement it themselves?
Reviewer 3 Report
Thankyou for giving me the possibility to review the paper "Developing a multilevel Decision Support Tool for urban mobility".
The paper aims to provide a comprehensive description of the interactive Decision Support Tool implemented to help cities and decision makers to design their strategy and shape the urban mobility of the future.
However before considering it for publication on "Sustainability" please discuss better the findings of the present study and draw a more detailed conclusion by highlighting the potential pratical implications in our cities.
Round 2
Reviewer 1 Report
The manuscript has been revised and some of the questions asked during the previous review have been answered. Regardless, I still think the current form needs better literature study, possibly not directly related to decision tools for urban mobility, but can be done separately for decision tools, and or urban mobility.
The authors also need to revised the reference style.
Author Response
Μanuscript has been revised and comments from previous review have been answered. Literature study sections has been enriched and the reference style has been revised